# Neural-Guided Deductive Search for Real-Time Program Synthesis from Examples

**Ashwin K. Vijayakumar**[*][†] **& Dhruv Batra**
School of Interactive Computing
Georgia Tech
Atlanta, GA 30308, USA
`{ashwinkv,dbatra}@gatech.edu`

**Abhishek Mohta**[†] **& Prateek Jain**
Microsoft Research India
Bengaluru, Karnataka 560001, India
`{t-abmoht,prajain}@microsoft.com`

**Oleksandr Polozov & Sumit Gulwani**
Microsoft Research Redmond
Redmond, WA 98052, USA
`{polozov,sumitg}@microsoft.com`

## Abstract

Synthesizing user-intended programs from a small number of input-output examples is a challenging problem with several important applications like spreadsheet manipulation, data wrangling and code refactoring. Existing synthesis systems either completely rely on deductive logic techniques that are extensively hand-engineered or on purely statistical models that need massive amounts of data, and in general fail to provide real-time synthesis on challenging benchmarks. In this work, we propose *Neural Guided Deductive Search* (NGDS), a hybrid synthesis technique that combines the best of both symbolic logic techniques and statistical models. Thus, it produces programs that satisfy the provided specifications by construction and generalize well on unseen examples, similar to data-driven systems. Our technique effectively utilizes the deductive search framework to reduce the learning problem of the neural component to a simple supervised learning setup. Further, this allows us to both train on sparingly available real-world data and still leverage powerful recurrent neural network encoders. We demonstrate the effectiveness of our method by evaluating on real-world customer scenarios by synthesizing accurate programs with up to $12\times$ speed-up compared to state-of-the-art systems.

## 1 Introduction

Automatic synthesis of programs that satisfy a given specification is a classical problem in AI (Waldinger & Lee, 1969), with extensive literature in both machine learning and programming languages communities. Recently, this area has gathered widespread interest, mainly spurred by the emergence of a sub-area – *Programming by Examples* (PBE) (Gulwani, 2011). A PBE system synthesizes programs that map a given set of example inputs to their specified example outputs. Such systems make many tasks accessible to a wider audience as example-based specifications can be easily provided even by end users without programming skills. See Figure 1 for an example. PBE systems are usually evaluated on three key criteria: **(a)** *correctness*: whether the synthesized program

| Input | Output |
|---|---|
| Yann LeCunn | Y LeCunn |
| Hugo Larochelle | H Larochelle |
| Tara Sainath | T Sainath |
| *Yoshua Bengio* | ? |

Figure 1: An example input-output spec; the goal is to learn a program that maps the given inputs to the corresponding outputs *and* generalizes well to new inputs. Both programs below satisfy the spec: **(i)** Concat(1st letter of 1st word, 2nd word), **(ii)** Concat(4th-last letter of 1st word, 2nd word). However, program **(i)** clearly generalizes better: for instance, its output on "Yoshua Bengio" is "Y Bengio" while program **(ii)** produces "s Bengio".

---

[*]Work done during an internship at Microsoft Research.

[†]Equal contribution.

satisfies the spec *i.e.* the provided example input-output mapping, **(b)** *generalization*: whether the program produces the desired outputs on *unseen* inputs, and finally, **(c)** *performance*: synthesis time.

State-of-the-art PBE systems are either *symbolic*, based on enumerative or deductive search (Gulwani, 2011; Polozov & Gulwani, 2015) or *statistical*, based on data-driven learning to induce the most likely program for the spec (Gaunt et al., 2016; Balog et al., 2017; Devlin et al., 2017). Symbolic systems are designed to produce a correct program *by construction* using logical reasoning and domain-specific knowledge. They also produce the *intended* program with few input-output examples (often just 1). However, they require significant engineering effort and their underlying search processes struggle with real-time performance, which is critical for user-facing PBE scenarios.

In contrast, statistical systems do not rely on specialized deductive algorithms, which makes their implementation and training easier. However, they lack in two critical aspects. First, they require a lot of training data and so are often trained using *randomly* generated tasks. As a result, induced programs can be fairly unnatural and fail to generalize to real-world tasks with a small number of examples. Second, purely statistical systems like RobustFill (Devlin et al., 2017) do not *guarantee* that the generated program satisfies the spec. Thus, solving the synthesis task requires generating multiple programs with a beam search and post-hoc filtering, which defeats real-time performance.

**Neural-Guided Deductive Search**   Motivated by shortcomings of both the above approaches, we propose *Neural-Guided Deductive Search* (NGDS), a hybrid synthesis technique that brings together the desirable aspects of both methods. The symbolic foundation of NGDS is *deductive search* (Polozov & Gulwani, 2015) and is parameterized by an underlying *domain-specific language* (DSL) of target programs. Synthesis proceeds by recursively applying production rules of the DSL to decompose the initial synthesis problem into *smaller* sub-problems and further applying the same search technique on them. Our **key observation I** is that most of the deduced sub-problems do not contribute to the final best program and therefore *a priori* predicting the usefulness of pursuing a particular sub-problem streamlines the search process resulting in considerable time savings. In NGDS, we use a statistical model trained on real-world data to predict a score that corresponds to the likelihood of finding a *generalizable* program as a result of exploring a sub-problem branch.

Our **key observation II** is that speeding up deductive search while retaining its correctness or generalization requires a close integration of symbolic and statistical approaches via an intelligent controller. It is based on the "branch & bound" technique from combinatorial optimization (Clausen, 1999). The overall algorithm integrates (i) deductive search, (ii) a statistical model that predicts, *a priori*, the generalization score of the best program from a branch, and (iii) a controller that selects sub-problems for further exploration based on the model's predictions.

Since program synthesis is a sequential process wherein a sequence of decisions (here, selections of DSL rules) collectively construct the final program, a reinforcement learning setup seems more natural. However, our **key observation III** is that deductive search is *Markovian* – it generates *independent* sub-problems at every level. In other words, we can reason about a satisfying program for the sub-problem without factoring in the bigger problem from which it was deduced. This brings three benefits enabling a *supervised learning* formulation: **(a)** a dataset of search decisions at every level over a relatively small set of PBE tasks that contains an exponential amount of information about the DSL promoting generalization, **(b)** such search traces can be generated and used for *offline* training, **(c)** we can learn separate models for different classes of sub-problems (e.g. DSL levels or rules), with relatively simpler supervised learning tasks.

**Evaluation**   We evaluate NGDS on the string transformation domain, building on top of PROSE, a commercially successful deductive synthesis framework for PBE (Polozov & Gulwani, 2015). It represents one of the most widespread and challenging applications of PBE and has shipped in multiple mass-market tools including Microsoft Excel and Azure ML Workbench.[1] We train and validate our method on 375 scenarios obtained from real-world customer tasks (Gulwani, 2011; Devlin et al., 2017). Thanks to the Markovian search properties described above, these scenarios generate a dataset of $400,000+$ intermediate search decisions. NGDS produces intended programs on $68\%$ of the scenarios despite using only *one* input-output example. In contrast, state-of-the-art neural synthesis techniques (Balog et al., 2017; Devlin et al., 2017) learn intended programs from a

---

[1]https://microsoft.github.io/prose/impact/

single example in only 24-36% of scenarios taking $\approx 4\times$ more time. Moreover, NGDS matches the accuracy of baseline PROSE while providing a speed-up of up to $12\times$ over challenging tasks.

**Contributions**  First, we present a branch-and-bound optimization based controller that exploits deep neural network based score predictions to select grammar rules efficiently (Section 3.2). Second, we propose a program synthesis algorithm that combines key traits of a symbolic and a statistical approach to retain desirable properties like correctness, robust generalization, and real-time performance (Section 3.3). Third, we evaluate NGDS against state-of-the-art baselines on real customer tasks and show significant gains (speed-up of up to $12\times$) on several critical cases (Section 4).

## 2  BACKGROUND

In this section, we provide a brief background on PBE and the PROSE framework, using established formalism from the programming languages community.

**Domain-Specific Language**  A program synthesis problem is defined over a *domain-specific language* (DSL). A DSL is a restricted programming language that is suitable for expressing tasks in a given domain, but small enough to restrict a search space for program synthesis. For instance, typical real-life DSLs with applications in textual data transformations (Gulwani, 2011) often include conditionals, limited forms of loops, and domain-specific operators such as string concatenation, regular expressions, and date/time formatting. DSLs for tree transformations such as code refactoring (Rolim et al., 2017) and data extraction (Le & Gulwani, 2014) include list/data-type processing operators such as Map and Filter, as well as domain-specific matching operators. Formally, a DSL $\mathcal{L}$ is specified as a context-free grammar, with each non-terminal symbol $N$ defined by a set of productions. The right-hand side of each production is an application of some operator $F(N_1, \ldots, N_k)$ to some symbols of $\mathcal{L}$. All symbols and operators are strongly typed. Figure 2 shows a subset of the Flash Fill DSL that we use as a running example in this paper.

**Inductive Program Synthesis**  The task of inductive program synthesis is characterized by a *spec*. A spec $\varphi$ is a set of $m$ input-output *constraints* $\{\sigma_i \rightsquigarrow \psi_i\}_{i=1}^m$, where:

- $\sigma$, an *input state* is a mapping of free variables of the desired program $P$ to some correspondingly typed values. At the top level of $\mathcal{L}$, a program (and its expected input state) has only one free variable – the *input variable* of the DSL (e.g., $inputs$ in Figure 2). Additional local variables are introduced inside $\mathcal{L}$ with a `let` construct.
- $\psi$ is an *output constraint* on the execution result of the desired program $P(\sigma_i)$. At the top level of $\mathcal{L}$, when provided by the user, $\psi$ is usually the *output example* – precisely the expected result of $P(\sigma_i)$. However, other intermediate constraints arise during the synthesis process. For instance, $\psi$ may be a *disjunction* of multiple allowed outputs.

The overall goal of program synthesis is thus: given a spec $\varphi$, find a program $P$ in the underlying DSL $\mathcal{L}$ that *satisfies* $\varphi$, *i.e.*, its outputs $P(\sigma_i)$ satisfy all the corresponding constraints $\psi_i$.

**Example 1.**  Consider the task of formatting a phone number, characterized by the spec $\varphi = \{inputs: [\text{``(612) 8729128''}]\} \rightsquigarrow \text{``612-872-9128''}$. It has a single input-output example, with an input state $\sigma$ containing a single variable $inputs$ and its value which is a list with a single input string. The output constraint is simply the desired program result.

The program the user is most likely looking for is the one that extracts (a) the part of the input enclosed in the first pair of parentheses, (b) the 7th to 4th characters from the end, and (c) the last 4 characters, and then concatenates all three parts using hyphens. In our DSL, this corresponds to:

$$\mathsf{Concat}\big(\mathsf{SubStr_0}(\mathsf{RegexPosition}(x, \langle\text{``(''}, \varepsilon\rangle, 0), \mathsf{RegexPosition}(x, \langle\varepsilon, \text{``)''}\rangle, 0)), \quad \mathsf{ConstStr}(\text{``-''}),$$
$$\mathsf{SubStr_0}(\mathsf{AbsolutePosition}(x, -8), \mathsf{AbsolutePosition}(x, -5)), \quad \mathsf{ConstStr}(\text{``-''}),$$
$$\mathsf{SubStr_0}(\mathsf{AbsolutePosition}(x, -5), \mathsf{AbsolutePosition}(x, -1)))$$

where $\varepsilon$ is an empty regex, $\mathsf{SubStr_0}(pos_1, pos_2)$ is an abbreviation for "`let` $x = \mathsf{std.Kth}(inputs, 0)$ `in` $\mathsf{Substring}(x, \langle pos_1, pos_2\rangle)$", and $\langle\cdot\rangle$ is an abbreviation for $\mathsf{std.Pair}$.

However, many other programs in the DSL also satisfy $\varphi$. For instance, all occurrences of "8" in the output can be produced via a subprogram that simply extracts the last character. Such a program overfits to $\varphi$ and is bound to fail for other inputs where the last character and the 4th one differ.

```
// Nonterminals
@start string transform ≔ atom | Concat(atom, transform);
string atom ≔ ConstStr(s)
            | let string x = std.Kth(inputs, k) in Substring(x, pp);
Tuple<int, int> pp ≔ std.Pair(pos, pos) | RegexOccurrence(x, r, k);
int pos ≔ AbsolutePosition(x, k) | RegexPosition(x, std.Pair(r, r), k);
// Terminals
@input string[] inputs;        string s;        int k;        Regex r;
```

Figure 2: A subset of the FlashFill DSL (Gulwani, 2011), used as a running example in this paper. Every program takes as input a list of strings $inputs$, and returns an output string, a *concatenation* of *atoms*. Each atom is either a constant or a substring of one of the inputs ($x$), extracted using some position logic. The RegexOccurrence position logic finds $k^{\text{th}}$ occurrence of a regex $r$ in $x$ and returns its boundaries. Alternatively, start and end positions can be selected independently either as absolute indices in $x$ from left or right (AbsolutePosition) or as the $k^{\text{th}}$ occurrence of a pair of regexes surrounding the position (RegexPosition). See Gulwani (2011) for an in-depth DSL description.

As Example 1 shows, typical real-life problems are severely underspecified. A DSL like FlashFill may contain up to $10^{20}$ programs that satisfy a given spec of 1-3 input-output examples (Polozov & Gulwani, 2015). Therefore, the main challenge lies in finding a program that not only satisfies the provided input-output examples but also generalizes to *unseen inputs*. Thus, the synthesis process usually interleaves *search* and *ranking*: the search phase finds a set of *spec-satisfying* programs in the DSL, from which the ranking phase selects top programs ordered using a domain-specific ranking function $h \colon \mathcal{L} \times \vec{\Sigma} \to \mathbb{R}$ where $\Sigma$ is the set of all input states. The ranking function takes as input a candidate program $P \in \mathcal{L}$ and a set of input states $\vec{\sigma} \in \vec{\Sigma}$ (usually $\vec{\sigma} =$ inputs in the given spec + any available unlabeled inputs), and produces a score for $P$'s *generalization*.

The implementation of $h$ expresses a subtle balance between program generality, complexity, and behavior on available inputs. For instance, in FlashFill $h$ penalizes overly specific regexes, prefers programs that produce fewer empty outputs, and prioritizes lower Kolmogorov complexity, among other features. In modern PBE systems like PROSE, $h$ is usually learned in a data-driven manner from customer tasks (Singh & Gulwani, 2015; Ellis & Gulwani, 2017). While designing and learning such a ranking is an interesting problem in itself, in this work we assume a black-box access to $h$. Finally, the problem of inductive program synthesis can be summarized as follows:

**Problem 1.** *Given a DSL $\mathcal{L}$, a ranking function $h$, a spec $\varphi = \{\sigma_i \rightsquigarrow \psi_i\}_{i=1}^{m}$, optionally a set of unlabeled inputs $\vec{\sigma}_u$, and a target number of programs $K$, let $\vec{\sigma} = \vec{\sigma}_u \cup \{\sigma_i\}_{i=1}^{m}$. The goal of* **inductive program synthesis** *is to find a program set $\mathcal{S} = \{P_1, \ldots, P_K\} \subset \mathcal{L}$ such that (a) every program in $\mathcal{S}$ satisfies $\varphi$, and (b) the programs in $\mathcal{S}$ generalize best: $h(P_i, \vec{\sigma}) \geq h(P, \vec{\sigma})$ for any other $P \in \mathcal{L}$ that satisfies $\varphi$.*

**Search Strategy**   Deductive search strategy for program synthesis, employed by PROSE explores the grammar of $\mathcal{L}$ *top-down* – iteratively unrolling the productions into partial programs starting from the root symbol. Following the divide-and-conquer paradigm, at each step it reduces its synthesis problem to smaller subproblems defined over the parameters of the current production. Formally, given a spec $\varphi$ and a symbol $N$, PROSE computes the set Learn($N, \varphi$) of top programs w.r.t. $h$ using two guiding principles:

1. If $N$ is defined through $n$ productions $N ≔ F_1(\ldots) \mid \ldots \mid F_n(\ldots)$, PROSE finds a $\varphi$-satisfying program set for *every* $F_i$, and unites the results, i.e., Learn($N, \varphi$) $= \cup_i$ Learn($F_i(\ldots), \varphi$).
2. For a given production $N ≔ F(N_1, \ldots, N_k)$, PROSE spawns off $k$ smaller synthesis problems Learn($N_j, \varphi_j$), $1 \leq j \leq k$ wherein PROSE deduces necessary and sufficient specs $\varphi_j$ for each $N_j$ such that every program of type $F(P_1, \ldots, P_k)$, where $P_j \in$ Learn($N_j, \varphi_j$), satisfies $\varphi$. The deduction logic (called a *witness function*) is domain-specific for each operator $F$. PROSE then again recursively solves each subproblem and unites a *cross-product* of the results.

**Example 2.** Consider a spec $\varphi = \{$"Yann" $\rightsquigarrow$ "Y.L"$\}$ on a $transform$ program. Via the first production $transform ≔ atom$, the only $\varphi$-satisfying program is ConstStr("Y.L"). The second production on the same level is Concat($atom, transform$). A necessary & sufficient spec on the $atom$ sub-program is that it should produce *some prefix* of the output string. Thus, the witness function for the Concat operator produces a *disjunctive spec* $\varphi_a = \{$"Yann" $\rightsquigarrow$ "Y" $\vee$ "Y."$\}$. Each

of these disjuncts, in turn, induces a corresponding necessary and sufficient *suffix* spec on the second parameter: $\varphi_{t1} = \{\text{"Yann"} \rightsquigarrow \text{".L"}\}$, and $\varphi_{t2} = \{\text{"Yann"} \rightsquigarrow \text{"L"}\}$, respectively. The disjuncts in $\varphi_a$ will be recursively satisfied by different program sets: "Y." can only be produced via an *atom* path with a ConstStr program, whereas "Y" can also be *extracted* from the input using many Substring logics (their generalization capabilities vary). Figure 3 shows the resulting search DAG.

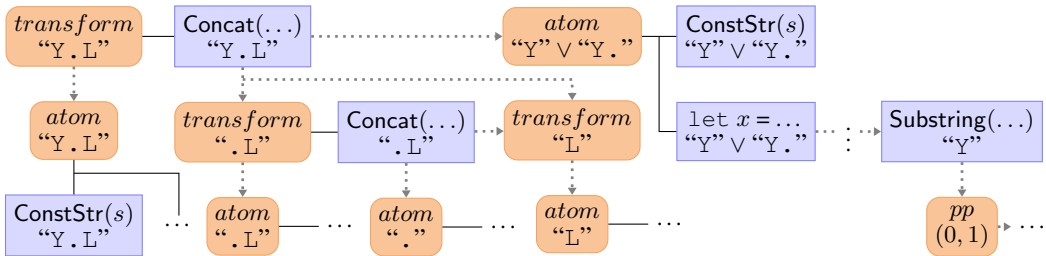

Figure 3: A portion of the search DAG from Example 2. Only the output parts of the respective specs are shown in each node, their common input state is a single string "Yann". Dashed arrows show recursive Learn calls on a corresponding DSL symbol.

Notice that the above mentioned principles create *logical non-determinism* due to which we might need to explore multiple alternatives in a search tree. As such non-determinism arises at every level of the DSL with potentially any operator, the search tree (and the resulting search process) is exponential in size. While all the branches of the tree by construction produce programs that satisfy the given spec, most of the branches do not contribute to the overall top-ranked *generalizable* program. During deductive search, PROSE has limited information about the programs potentially produced from each branch, and cannot estimate their quality, thus exploring the entire tree unnecessarily. Our main contribution is a *neural-guided search algorithm* that predicts the best program scores from each branch, and allows PROSE to omit branches that are unlikely to produce the desired program *a priori*.

## 3 SYNTHESIS ALGORITHM

Consider an arbitrary branching moment in the top-down search strategy of PROSE. For example, let $N$ be a nonterminal symbol in $\mathcal{L}$, defined through a set of productions $N := F_1(\ldots) \mid \ldots \mid F_n(\ldots)$, and let $\varphi$ be a spec on $N$, constructed earlier during the recursive descent over $\mathcal{L}$. A conservative way to select the top $k$ programs rooted at $N$ (as defined by the ranking function $h$), i.e., to compute $\text{Learn}(N, \varphi)$, is to learn the top $k$ programs of kind $F_i(\ldots)$ for all $i \in [k]$ and then select the top $k$ programs overall from the union of program sets learned for each production. Naturally, exploring all the branches for each nonterminal in the search tree is computationally expensive.

In this work, we propose a data-driven method to select an appropriate production rule $N := F_i(N_1, \ldots, N_k)$ that would most likely lead to a top-ranked program. To this end, we use the current spec $\varphi$ to determine the "optimal" rule. Now, it might seem unintuitive that even without exploring a production rule and finding the best program in the corresponding program set, we can *a priori* determine optimality of that rule. However, we argue that by understanding $\varphi$ and its relationship with the ranking function $h$, we can *predict* the intended branch in many real-life scenarios.

**Example 3.** Consider a spec $\varphi = \{\text{"alice"} \rightsquigarrow \text{"alice@iclr.org"}, \text{"bob"} \rightsquigarrow \text{"bob@iclr.org"}\}$. While learning a program in $\mathcal{L}$ given by Figure 2 that satisfies $\varphi$, it is clear right at the beginning of the search procedure that the rule $transform := atom$ does not apply. This is because any programs derived from $transform := atom$ can either extract a substring from the input or return a constant string, both of which fail to produce the desired output. Hence, we should only consider $transform := \text{Concat}(\ldots)$, thus significantly reducing the search space.

Similarly, consider another spec $\varphi = \{\text{"alice smith"} \rightsquigarrow \text{"alice"}, \text{"bob jones"} \rightsquigarrow \text{"bob"}\}$. In this case, the output appears to be a substring of input, thus selecting $transform := atom$ at the beginning of the search procedure is a better option than $transform := \text{Concat}(\ldots)$.

However, many such decisions are more subtle and depend on the ranking function $h$ itself. For example, consider a spec $\varphi = \{\text{"alice liddell"} \rightsquigarrow \text{"al"}, \text{"bob ong"} \rightsquigarrow \text{"bo"}\}$. Now,

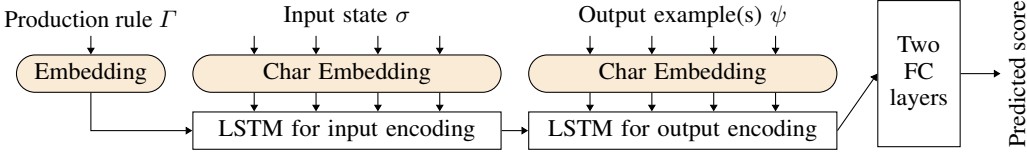

Figure 4: LSTM-based model for predicting the score of a candidate production for a given spec $\varphi$.

both $transform \coloneqq atom$ and $transform \coloneqq \mathsf{Concat}(\ldots)$ may lead to viable programs because the output can be constructed using the first two letters of the input (i.e. a substring atom) or by *concatenating* the first letters of each word. Hence, the branch that produces the best program is ultimately determined by the ranking function $h$ since both branches generate valid programs.

Example 3 shows that to design a data-driven search strategy for branch selection, we need to learn the subtle relationship between $\varphi$, $h$, and the candidate branch. Below, we provide one such model.

### 3.1 Predicting the Generalization Score

As mentioned above, our goal is to predict one or more production rules that for a given spec $\varphi$ will lead to a top-ranked program (as ranked *a posteriori* by $h$). Formally, given black-box access to $h$, we want to learn a function $f$ such that,

$$f(\Gamma, \varphi) \approx \max_{P \in \mathcal{S}(\Gamma, \varphi)} h(P, \varphi),$$

where $\Gamma$ is a production rule in $\mathcal{L}$, and $\mathcal{S}(\Gamma, \varphi)$ is a *program set* of all DSL programs derived from the rule $\Gamma$ that satisfy $\varphi$. In other words, we want to predict the score of the top-ranked $\varphi$-*satisfying* program that is synthesized by unrolling the rule $\Gamma$. We assume that the symbolic search of PROSE handles the construction of $\mathcal{S}(\Gamma, \varphi)$ and ensures that programs in it satisfy $\varphi$ by construction. The goal of $f$ is to optimize the score of a program derived from $\Gamma$ *assuming* this program is valid. If no program derived from $\Gamma$ can satisfy $\varphi$, $f$ should return $-\infty$. Note that, drawing upon observations mentioned in Section 1, we have cast the production selection problem as a *supervised learning* problem, thus simplifying the learning task as opposed to end-to-end reinforcement learning solution.

We have evaluated two models for learning $f$. The loss function for the prediction is given by:

$$L(f; \Gamma, \varphi) = \left(f(\Gamma, \varphi) - \max_{P \in \mathcal{S}(\Gamma, \varphi)} h(P, \varphi)\right)^2.$$

Figure 4 shows a common structure of both models we have evaluated. Both are based on a standard multi-layer LSTM architecture (Hochreiter & Schmidhuber, 1997) and involve **(a)** embedding the given spec $\varphi$, **(b)** encoding the given production rule $\Gamma$, and **(c)** a feed-forward network to output a score $f(\Gamma, \varphi)$. One model attends over input when it encodes the output, whereas another does not.

### 3.2 Controller for Branch Selection

A score model $f$ alone is insufficient to perfectly predict the branches that should be explored at every level. Consider again a branching decision moment $N \coloneqq F_1(\ldots) \mid \ldots \mid F_n(\ldots)$ in a search process for top $k$ programs satisfying a spec $\varphi$. One naïve approach to using the predictions of $f$ is to always follow the highest-scored production rule $\arg\max_i f(F_i, \varphi)$. However, this means that *any single incorrect decision on the path from the DSL root to the desired program will eliminate that program from the learned program set*. If our search algorithm fails to produce the desired program by committing to a suboptimal branch anytime during the search process, then the user may never discover that such a program exists unless they supply additional input-output example.

Thus, a branch selection strategy based on the predictions of $f$ must balance a trade-off of *performance* and *generalization*. Selecting too few branches (a single best branch in the extreme case) risks committing to an incorrect path early in the search process and producing a suboptimal program or no program at all. Selecting too many branches (all $n$ branches in the extreme case) is no different from baseline PROSE and fails to exploit the predictions of $f$ to improve its performance.

Formally, a *controller* for branch selection at a symbol $N \coloneqq F_1(\ldots) \mid \ldots \mid F_n(\ldots)$ targeting $k$ best programs must **(a)** predict the expected score of the best program from each program set:

**function** THRESHOLDBASED($\varphi, h, k, s_1, \ldots, s_n$)
1:   Result set $\mathcal{S}^* \leftarrow []$
2:   $i^* \leftarrow \operatorname{argmax}_i s_i$
3:   **for all** $1 \leq i \leq n$ **do**
4:     **if** $|s_i - s_{i^*}| \leq \theta$ **then**
        // *Recursive search*
5:         $\mathcal{S}^* \mathrel{+}= \text{LEARN}(F_i, \varphi, k)$
6:   **return** the top $k$ programs of $\mathcal{S}$ w.r.t. $h$

**function** BNBBASED($\varphi, h, k, s_1, \ldots, s_n$)
1:   Result set $\mathcal{S}^* \leftarrow []$;    Program target $k' \leftarrow k$
2:   Reorder $F_i$ in the descending order of $s_i$
3:   **for all** $1 \leq i \leq n$ **do**
4:     $\mathcal{S}_i \leftarrow \text{LEARN}(F_i, \varphi, k')$   // *Recursive search*
5:     $j \leftarrow \text{BINARYSEARCH}(s_{i+1}, \text{Map}(h, \mathcal{S}_i))$
6:     $\mathcal{S}^* = \mathcal{S}_i^* \cup \mathcal{S}_i[0..j]$;   $k' \leftarrow k' - j$
7:     **if** $k' \leq 0$ **then break**
8:   **return** $\mathcal{S}^*$

Figure 5: The controllers for guiding the search process to construct a *most generalizable* $\varphi$-satisfying program set $\mathcal{S}$ of size $k$ given the $f$-predicted best scores $s_1, \ldots, s_n$ of the productions $F_1, \ldots, F_n$.

**Given:**   DSL $\mathcal{L}$, ranking function $h$, controller $\mathcal{C}$ from Figure 5 (THRESHOLDBASED or BNBBASED), symbolic search algorithm LEARN(Production rule $\Gamma$, spec $\varphi$, target $k$) as in PROSE (Polozov & Gulwani, 2015, Figure 7) with all recursive calls to LEARN replaced with LEARNNGDS

**function** LEARNNGDS(Symbol $N := F_1(\ldots) \mid \ldots \mid F_n(\ldots)$, spec $\varphi$, target number of programs $k$)
1:   **if** $n = 1$ **then return** LEARN($F_1, \varphi, k$)
2:   Pick a score model $f$ based on $\text{depth}(N, \mathcal{L})$
3:   $s_1, \ldots, s_n \leftarrow f(F_1, \varphi), \ldots, f(F_n, \varphi)$
4:   **return** $\mathcal{C}(\varphi, h, k, s_1, \ldots, s_n)$

Figure 6: Neural-guided deductive search over $\mathcal{L}$, parameterized with a branch selection controller $\mathcal{C}$.

$s_i = f(F_i, \varphi) \; \forall 1 \leq i \leq n$, and **(b)** use the predicted scores $s_i$ to narrow down the set of productions $F_1, \ldots, F_n$ to explore and to obtain the overall result by selecting a subset of generated programs. In this work, we propose and evaluate two controllers. Their pseudocode is shown in Figure 5.

**Threshold-based:** Fix a *score threshold* $\theta$, and explore those branches whose predicted score differs by at most $\theta$ from the maximum predicted score. This is a simple extension of the naïve "argmax" controller discussed earlier that also explores any branches that are predicted "approximately as good as the best one". When $\theta = 0$, it reduces to the "argmax" one.

**Branch & Bound:** This controller is based on the "branch & bound" technique in combinatorial optimization (Clausen, 1999). Assume the branches $F_i$ are ordered in the descending order of their respective predicted scores $s_i$. After recursive learning produces its program set $\mathcal{S}_i$, the controller proceeds to the next branch only if $s_{i+1}$ *exceeds the score of the worst program in* $\mathcal{S}_i$. Moreover, it reduces the target number of programs to be learned, using $s_{i+1}$ as a lower bound on the scores of the programs in $\mathcal{S}_i$. That is, rather than relying blindly on the predicted scores, the controller guides the remaining search process by accounting for the actual synthesized programs as well.

## 3.3   NEURAL-GUIDED DEDUCTIVE SEARCH

We now combine the above components to present our unified algorithm for program synthesis. It builds upon the *deductive search* of the PROSE system, which uses symbolic PL insights in the form of *witness functions* to construct and narrow down the search space, and a *ranking function* $h$ to pick the most generalizable program from the found set of spec-satisfying ones. However, it significantly speeds up the search process by guiding it *a priori* at each branching decision using the learned score model $f$ and a branch selection controller, outlined in Sections 3.1 and 3.2. The resulting *neural-guided deductive search* (NGDS) keeps the symbolic insights that construct the search tree ensuring correctness of the found programs, but explores only those branches of this tree that are likely to produce the user-intended generalizable program, thus eliminating unproductive search time.

A key idea in NGDS is that the score prediction model $f$ does not have to be the same for all decisions in the search process. It is possible to train separate models for different DSL levels, symbols, or even productions. This allows the model to use different features of the input-output spec for evaluating the fitness of different productions, and also leads to much simpler supervised learning problems.

Figure 6 shows the pseudocode of NGDS. It builds upon the deductive search of PROSE, but augments every branching decision on a symbol with some branch selection controller from Section 3.2. We present a comprehensive evaluation of different strategies in Section 4.

| Metric | PROSE | $DC_1$ | $DC_2$ | $DC_3$ | $RF_1$ | $RF_2$ | $RF_3$ | NGDS |
|---|---|---|---|---|---|---|---|---|
| **Accuracy (% of 73)** | 67.12 | 35.81 | 47.38 | 62.92 | 24.53 | 39.72 | 56.41 | **68.49** |
| **Speed-up ($\times$ PROSE)** | 1.00 | **1.82** | 1.53 | 1.42 | 0.25 | 0.27 | 0.30 | 1.67 |

Table 1: Accuracy and average speed-up of NGDS vs. baseline methods. Accuracies are computed on a test set of 73 tasks. *Speed-up* of a method is the geometric mean of its per-task speed-up (ratio of synthesis time of PROSE and of the method) when restricted to a subset of tasks with PROSE's synthesis time is $\geq 0.5$ sec.

## 4 EVALUATION

In this section, we evaluate our NGDS algorithm over the string manipulation domain with a DSL given by Figure 2; see Figure 1 for an example task. We evaluate NGDS, its ablations, and baseline techniques on two key metrics: (a) generalization accuracy on unseen inputs, (b) synthesis time.

**Dataset.** We use a dataset of 375 *tasks* collected from real-world customer string manipulation problems, split into 65% training, 15% validation, and 20% test data. Some of the common applications found in our dataset include date/time formatting, manipulating addresses, modifying names, automatically generating email IDs, etc. Each task contains about 10 inputs, of which *only one* is provided as the spec to the synthesis system, mimicking industrial applications. The remaining *unseen* examples are used to evaluate generalization performance of the synthesized programs. After running synthesis of top-1 programs with PROSE on all training tasks, we have collected a dataset of $\approx 400,000$ intermediate search decisions, *i.e.* triples $\langle$production $\Gamma$, spec $\varphi$, *a posteriori* best score $h(P, \varphi)\rangle$.

**Baselines.** We compare our method against two state-of-the-art neural synthesis algorithms: RobustFill (Devlin et al., 2017) and DeepCoder (Balog et al., 2017). For RobustFill, we use the best-performing *Attention-C* model and use their recommended DP-Beam Search with a beam size of 100 as it seems to perform the best; Table 3 in Appendix A presents results with different beam sizes. As in the original work, we select the top-1 program ranked according to the generated $\log$-likelihood. DeepCoder is a generic framework that allows their neural predictions to be combined with any program synthesis method. So, for fair comparison, we combine DeepCoder's predictions with PROSE. We train DeepCoder model to predict a distribution over $\mathcal{L}$'s operators and as proposed, use it to guide PROSE synthesis. Since both RobustFill and DeepCoder are trained on randomly sampled programs and are not optimized for generalization in the real-world, we include their variants trained with 2 or 3 examples (denoted $RF_m$ and $DC_m$) for fairness, although $m = 1$ example is the most important scenario in real-life industrial usage.

**Ablations.** As mentioned in Section 3, our novel usage of score predictors to guide the search enables us to have multiple prediction models and controllers at various stages of the synthesis process. Here we investigate ablations of our approach with models that specialize in predictions for individual levels in the search process. The model $T_1$ is trained for symbol $transform$ (Figure 2) when expanded in the first level. Similarly, $PP$, $POS$ refer to models trained for the $pp$ and $pos$ symbol, respectively. Finally, we train all our LSTM-based models with CNTK (Seide & Agarwal, 2016) using Adam (Kingma & Ba, 2014) with a learning rate of $10^{-2}$ and a batch size of 32, using early stopping on the validation loss to select the best performing model (thus, 100-600 epochs).

We also evaluate three controllers: threshold-based (Thr) and branch-and-bound (BB) controllers given in Figure 5, and a combination of them – branch-and-bound with a 0.2 threshold predecessor ($BB_{0.2}$). In Tables 1 and 2 we denote different model combinations as NGDS($f, \mathcal{C}$) where $f$ is a symbol-based model and $\mathcal{C}$ is a controller. The final algorithm selection depends on its accuracy-performance trade-off. In Table 1, we use NGDS($T_1 + POS$, BB), the best performing algorithm on the test set, although NGDS($T_1$, BB) performs slightly better on the validation set.

**Evaluation Metrics.** *Generalization accuracy* is the percentage of test tasks for which the generated program satisfies *all* unseen inputs in the task. *Synthesis time* is measured as the wall-clock time taken by a synthesis method to find the correct program, median over 5 runs. We run all the methods on the same machine with 2.3 GHz Intel Xeon processor, 64GB of RAM, and Windows Server 2016.

**Results.** Table 1 presents generalization accuracy as well as synthesis time speed-up of various methods w.r.t. PROSE. As we strive to provide real-time synthesis, we only compare the times for tasks which require PROSE more than 0.5 sec. Note that, with one example, NGDS and PROSE are

| Method | Validation | | Test | | % of branches |
|---|---|---|---|---|---|
| | Accuracy | Speed-up | Accuracy | Speed-up | |
| PROSE | 70.21 | 1 | 67.12 | 1 | 100.00 |
| NGDS($T_1$, Thr) | 59.57 | 1.15 | 67.12 | 1.27 | 62.72 |
| NGDS($T_1$, BB) | 63.83 | 1.58 | 68.49 | 1.22 | 51.78 |
| NGDS($T_1$, BB$_{0.2}$) | 61.70 | 1.03 | 67.12 | 1.22 | 63.16 |
| NGDS($T_1 + PP$, Thr) | 59.57 | 0.76 | 67.12 | 0.97 | 56.41 |
| NGDS($T_1 + PP$, BB) | 61.70 | 1.05 | 72.60 | 0.89 | 50.22 |
| NGDS($T_1 + PP$, BB$_{0.2}$) | 61.70 | 0.72 | 67.12 | 0.86 | 56.43 |
| NGDS($T_1 + POS$, Thr) | 61.70 | 1.19 | 67.12 | 1.93 | 55.63 |
| NGDS($T_1 + POS$, BB) | 63.83 | 1.13 | 68.49 | 1.67 | 50.44 |
| NGDS($T_1 + POS$, BB$_{0.2}$) | 63.83 | 1.19 | 67.12 | 1.73 | 55.73 |

Table 2: Accuracies, mean speed-ups, and % of branches taken for different ablations of NGDS.

significantly more accurate than RobustFill and DeepCoder. This is natural as those methods are not trained to optimize generalization, but it also highlights advantage of a close integration with a symbolic system (PROSE) that incorporates deep domain knowledge. Moreover, on an average, our method saves more than $50\%$ of synthesis time over PROSE. While DeepCoder with one example speeds up the synthesis even more, it does so at the expense of accuracy, eliminating branches with *correct* programs in $65\%$ of tasks.

Table 2 presents speed-up obtained by variations of our models and controllers. In addition to generalization accuracy and synthesis speed-up, we also show a fraction of branches that were selected for exploration by the controller. Our method obtains impressive speed-up of $> 1.5\times$ in 22 cases. One such test case where we obtain $12\times$ speedup is a simple extraction case which is fairly common in Web mining: {"alpha,beta,charlie,delta" ⇝ "alpha"}. For such cases, our model determine $transform := atom$ to be the correct branch (that leads to the final Substring based program) and hence saves time required to explore the entire Concat operator which is expensive. Another interesting test case where we observe $2.7\times$ speed-up is: {"457 124th St S, Seattle, WA 98111" ⇝ "Seattle-WA"}. This test case involves learning a Concat operator initially followed by Substring and RegexPosition operator. Appendix B includes a comprehensive table of NGDS performance on all the validation and test tasks.

All the models in Table 2 run *without* attention. As measured by *score flip accuracies* (*i.e.* percentage of correct orderings of branch scores on the same level), attention-based models perform best, achieving $99.57/90.4/96.4\%$ accuracy on train/validation/test, respectively (as compared to $96.09/91.24/91.12\%$ for non-attention models). However, an attention-based model is significantly more computationally expensive at prediction time. Evaluating it dominates the synthesis time and eliminates any potential speed-ups. Thus, we decided to forgo attention in initial NGDS and investigate model compression/binarization in future work.

**Error Analysis.** As Appendix B shows, NGDS is slower than PROSE on some tasks. This occurs when the predictions do not satisfy the constraints of the controller *i.e.* all the predicted scores are within the threshold or they violate the actual scores during B&B exploration. This leads to NGDS evaluating the LSTM for branches that were previously pruned. This is especially harmful when branches pruned out at the very beginning of the search need to be reconsidered – as it could lead to evaluating the neural network many times. While a single evaluation of the network is quick, a search tree involves many evaluations, and when performance of PROSE is already $< 1$ s, this results in considerable *relative* slowdown. We provide two examples to illustrate both the failure modes:

**(a)** "41.7114830017,-91.41233825683,41.60762786865,-91.63739013671" ⇝ "41.7114830017". The intended program is a simple substring extraction. However, at depth 1, the predicted score of Concat is much higher than the predicted score of Atom, and thus NGDS explores only the Concat branch. The found Concat program is incorrect because it uses absolute position indexes and does not generalize to other similar extraction tasks. We found this scenario common with punctuation in the output string, which the model considers a strong signal for Concat.

**(b)** "type size = 36: Bartok.Analysis.CallGraphNode type size = 32: Bartok.Analysis.CallGraphNode CallGraphNode" ⇝ "36->32". In this case, NGDS correctly explores only the Concat branch, but the slowdown happens at the *pos* symbol.

There are many different logics to extract the "36" and "32" substrings. NGDS explores the RelativePosition branch first, but the score of the resulting program is less then the prediction for RegexPositionRelative. Thus, the B&B controller explores both branches anyway, which leads to a relative slowdown caused by the network evaluation time.

## 5 RELATED WORK

**Neural Program Induction** systems synthesize a program by training a *new* neural network model to map the example inputs to example outputs (Graves et al., 2014; Reed & De Freitas, 2016; Zaremba et al., 2016). Examples include Neural Turing Machines (Graves et al., 2014) that can learn simple programs like copying/sorting, work of Kaiser & Sutskever (2015) that can perform more complex computations like binary multiplications, and more recent work of Cai et al. (2017) that can incorporate recursions. While we are interested in ultimately producing the right output, all these models need to be re-trained for a given problem type, thus making them unsuitable for real-life synthesis of *different* programs with *few* examples.

**Neural Program Synthesis** systems synthesize a program in a given $\mathcal{L}$ with a pre-learned neural network. Seminal works of Bosnjak et al. (2017) and Gaunt et al. (2016) proposed first producing a high-level sketch of the program using procedural knowledge, and then synthesizing the program by combining the sketch with a neural or enumerative synthesis engine. In contrast, R3NN (Parisotto et al., 2016) and RobustFill (Devlin et al., 2017) systems synthesize the program end-to-end using a neural network; Devlin et al. (2017) show that RobustFill in fact outperforms R3NN. However, RobustFill does not guarantee generation of *spec-satisfying* programs and often requires more than one example to find the intended program. In fact, our empirical evaluation (Section 4) shows that our hybrid synthesis approach significantly outperforms the purely statistical approach of RobustFill.

DeepCoder (Balog et al., 2017) is also a hybrid synthesis system that guides enumerative program synthesis by prioritizing DSL operators according to a spec-driven likelihood distribution on the same. However, NGDS differs from DeepCoder in two important ways: (a) it guides the search process *at each recursive level* in a top-down *goal-oriented* enumeration and thus reshapes the search tree, (b) it is trained on real-world data instead of random programs, thus achieving better generalization.

**Symbolic Program Synthesis** has been studied extensively in the PL community (Gulwani et al., 2017; Alur et al., 2013), dating back as far as 1960s (Waldinger & Lee, 1969). Most approaches employ either bottom-up enumerative search (Udupa et al., 2013), constraint solving (Torlak & Bodik, 2013), or inductive logic programming (Lin et al., 2014), and thus scale poorly to real-world industrial applications (e.g. data wrangling applications). In this work, we build upon deductive search, first studied for synthesis by Manna & Waldinger (1971), and primarily used for program synthesis from formal logical specifications (Puschel et al., 2005; Chaudhari & Damani, 2015). Gulwani (2011) and later Polozov & Gulwani (2015) used it to build PROSE, a commercially successful domain-agnostic system for PBE. While its deductive search guarantees program correctness and also good generalization via an accurate ranking function, it still takes several seconds on complex tasks. Thus, speeding up deductive search requires considerable engineering to develop manual heuristics. NGDS instead integrates neural-driven predictions at each level of deductive search to alleviate this drawback. Work of Loos et al. (2017) represents the closest work with a similar technique but their work is applied to an automated theorem prover, and hence need not care about generalization. In contrast, NGDS guides the search toward generalizable programs while relying on the underlying symbolic engine to generate correct programs.

## 6 CONCLUSION

We studied the problem of real-time program synthesis with a small number of input-output examples. For this problem, we proposed a neural-guided system that builds upon PROSE, a state-of-the-art symbolic logic based system. Our system avoids top-down *enumerative* grammar exploration required by PROSE thus providing impressive synthesis performance while still retaining key advantages of a deductive system. That is, compared to existing neural synthesis techniques, our system enjoys following advantages: a) *correctness*: programs generated by our system are guaranteed to satisfy the given input-output specification, b) *generalization*: our system learns the user-intended program with just one input-output example in around 60% test cases while existing neural systems learn such a

program in only 16% test cases, c) *synthesis time*: our system can solve most of the test cases in less than 0.1 sec and provide impressive performance gains over both neural as well symbolic systems.

The key take-home message of this work is that a deep integration of a symbolic deductive inference based system with statistical techniques leads to best of both the worlds where we can avoid extensive engineering effort required by symbolic systems without compromising the quality of generated programs, and at the same time provide significant performance (when measured as synthesis time) gains. For future work, exploring better learning models for production rule selection and applying our technique to diverse and more powerful grammars should be important research directions.

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

## A ROBUSTFILL PERFORMANCE WITH DIFFERENT BEAM SIZES

For our experiments, we implemented RobustFill with the beam size of 100, as it presented a good trade-off between generalization accuracy and performance hit. The following table shows a detailed comparison of RobustFill's generalization accuracy and performance for different beam sizes and numbers of training examples.

| Number of examples ($m$) | Beam size | Accuracy (%) | Speed-up ($\times$ PROSE) |
|---|---|---|---|
| **1** | 10 | 18.4 | 0.41 |
| | 100 | 24.5 | 0.25 |
| | 1000 | 34.1 | 0.04 |
| **2** | 10 | 32.2 | 0.43 |
| | 100 | 39.7 | 0.27 |
| | 1000 | 47.6 | 0.04 |
| **3** | 10 | 49.8 | 0.48 |
| | 100 | 56.4 | 0.30 |
| | 1000 | 63.4 | 0.04 |

Table 3: Generalization accuracy and performance of RobustFill for different beam sizes and numbers of training examples.

## B PERFORMANCE OF BEST NGDS MODEL ON ALL NON-TRAINING TASKS

| Task # | Test/Val | PROSE Time (s) | NGDS Time (s) | Speed-up | PROSE Correct? | NGDS Correct? |
|---|---|---|---|---|---|---|
| 1 | Test | 3.0032564 | 0.233686 | 12.85167 | ✓ | ✓ |
| 2 | Validation | 1.1687841 | 0.211069 | 5.53745 | ✓ | ✗ |
| 3 | Validation | 0.4490832 | 0.1307367 | 3.43502 | ✓ | ✓ |
| 4 | Test | 6.665234 | 2.012157 | 3.312482 | ✓ | ✗ |
| 5 | Test | 2.28298 | 0.83715 | 2.727086 | ✗ | ✗ |
| 6 | Test | 3.0391034 | 1.1410092 | 2.663522 | ✓ | ✗ |
| 7 | Validation | 0.5487662 | 0.2105728 | 2.606064 | ✓ | ✓ |
| 8 | Test | 2.4120103 | 0.9588959 | 2.515404 | ✗ | ✗ |
| 9 | Validation | 7.6010733 | 3.052303 | 2.490275 | ✗ | ✗ |
| 10 | Test | 2.1165486 | 0.8816776 | 2.400592 | ✗ | ✗ |
| 11 | Test | 0.9622929 | 0.405093 | 2.375486 | ✓ | ✓ |
| 12 | Validation | 0.4033455 | 0.1936532 | 2.082824 | ✗ | ✗ |
| 13 | Test | 0.4012993 | 0.1929299 | 2.080026 | ✓ | ✓ |
| 14 | Validation | 2.9467418 | 1.4314372 | 2.05859 | ✓ | ✓ |
| 15 | Test | 0.3855433 | 0.1987497 | 1.939843 | ✗ | ✗ |
| 16 | Test | 6.0043011 | 3.1862577 | 1.884437 | ✗ | ✗ |
| 17 | Test | 3.0316721 | 1.6633142 | 1.82267 | ✗ | ✗ |
| 18 | Test | 0.3414629 | 0.1933263 | 1.766252 | ✓ | ✓ |
| 19 | Validation | 0.3454594 | 0.2014236 | 1.715089 | ✓ | ✓ |
| 20 | Test | 0.3185586 | 0.202928 | 1.569811 | ✗ | ✗ |
| 21 | Test | 0.2709963 | 0.1734634 | 1.562268 | ✓ | ✓ |
| 22 | Test | 0.4859534 | 0.3169533 | 1.533202 | ✓ | ✓ |
| 23 | Test | 0.8672071 | 0.5865048 | 1.478602 | ✓ | ✗ |
| 24 | Validation | 0.3626161 | 0.2590434 | 1.399828 | ✓ | ✓ |
| 25 | Validation | 2.3343791 | 1.6800684 | 1.389455 | ✓ | ✓ |
| 26 | Test | 0.2310051 | 0.1718745 | 1.344034 | ✓ | ✓ |
| 27 | Test | 0.1950921 | 0.1456817 | 1.339167 | ✓ | ✓ |
| 28 | Test | 0.8475303 | 0.6425532 | 1.319004 | ✓ | ✓ |
| 29 | Validation | 0.4064375 | 0.316499 | 1.284167 | ✓ | ✓ |
| 30 | Test | 0.2601689 | 0.2083826 | 1.248515 | ✗ | ✗ |
| 31 | Test | 0.2097732 | 0.1753706 | 1.196171 | ✓ | ✓ |
| 32 | Test | 1.2224533 | 1.0264273 | 1.190979 | ✗ | ✗ |
| 33 | Test | 0.5431827 | 0.4691296 | 1.157852 | ✓ | ✓ |
| 34 | Validation | 0.4183223 | 0.3685321 | 1.135104 | ✓ | ✓ |

| Task # | Test/Val | PROSE Time (s) | NGDS Time (s) | Speed-up | PROSE Correct? | NGDS Correct? |
|---|---|---|---|---|---|---|
| 35 | Validation | 0.2497723 | 0.2214195 | 1.12805 | ✗ | ✓ |
| 36 | Validation | 0.2385918 | 0.212407 | 1.123277 | ✗ | ✗ |
| 37 | Test | 0.2241414 | 0.2004937 | 1.117947 | ✓ | ✓ |
| 38 | Validation | 0.2079995 | 0.1880859 | 1.105875 | ✓ | ✓ |
| 39 | Test | 0.2788713 | 0.2654384 | 1.050606 | ✓ | ✓ |
| 40 | Test | 0.1821743 | 0.1758255 | 1.036109 | ✓ | ✓ |
| 41 | Validation | 0.1486939 | 0.1456755 | 1.02072 | ✓ | ✓ |
| 42 | Test | 0.3981185 | 0.3900767 | 1.020616 | ✗ | ✓ |
| 43 | Test | 0.9959218 | 0.9960901 | 0.999831 | ✓ | ✓ |
| 44 | Test | 0.2174055 | 0.2239088 | 0.970956 | ✓ | ✓ |
| 45 | Test | 1.8684116 | 1.9473475 | 0.959465 | ✓ | ✓ |
| 46 | Test | 0.1357812 | 0.1428591 | 0.950455 | ✓ | ✓ |
| 47 | Validation | 0.2549691 | 0.2709866 | 0.940892 | ✗ | ✗ |
| 48 | Test | 0.1650636 | 0.1762617 | 0.936469 | ✓ | ✓ |
| 49 | Validation | 0.5368683 | 0.5781537 | 0.928591 | ✓ | ✓ |
| 50 | Test | 0.1640937 | 0.1851361 | 0.886341 | ✗ | ✗ |
| 51 | Validation | 0.5006552 | 0.5736976 | 0.872681 | ✓ | ✓ |
| 52 | Test | 0.2064185 | 0.2401594 | 0.859506 | ✓ | ✓ |
| 53 | Validation | 0.2381335 | 0.277788 | 0.857249 | ✗ | ✗ |
| 54 | Test | 0.2171637 | 0.2677121 | 0.811184 | ✓ | ✓ |
| 55 | Test | 0.6307356 | 0.7807711 | 0.807837 | ✓ | ✓ |
| 56 | Validation | 0.3462029 | 0.4325302 | 0.800413 | ✓ | ✓ |
| 57 | Test | 0.4285604 | 0.5464594 | 0.784249 | ✗ | ✗ |
| 58 | Validation | 0.155915 | 0.1992245 | 0.78261 | ✓ | ✓ |
| 59 | Test | 0.1651815 | 0.2135129 | 0.773637 | ✗ | ✗ |
| 60 | Validation | 0.1212689 | 0.1571558 | 0.771648 | ✓ | ✓ |
| 61 | Test | 0.1980844 | 0.257616 | 0.768913 | ✓ | ✓ |
| 62 | Validation | 0.1534717 | 0.2004651 | 0.765578 | ✓ | ✓ |
| 63 | Test | 0.2443636 | 0.3258476 | 0.749932 | ✗ | ✗ |
| 64 | Test | 0.1217696 | 0.1635984 | 0.74432 | ✓ | ✓ |
| 65 | Validation | 0.2446501 | 0.3301224 | 0.741089 | ✓ | ✓ |
| 66 | Validation | 0.6579789 | 0.8886647 | 0.740413 | ✓ | ✗ |
| 67 | Test | 0.1490806 | 0.2022204 | 0.737218 | ✓ | ✓ |
| 68 | Test | 0.2668753 | 0.3681659 | 0.724878 | ✓ | ✓ |
| 69 | Test | 0.1072814 | 0.1487589 | 0.721176 | ✓ | ✓ |
| 70 | Validation | 0.1310034 | 0.181912 | 0.720147 | ✓ | ✗ |
| 71 | Test | 0.1954476 | 0.273414 | 0.714841 | ✓ | ✓ |
| 72 | Test | 0.3323319 | 0.468445 | 0.709436 | ✓ | ✓ |
| 73 | Test | 0.2679471 | 0.3806013 | 0.70401 | ✓ | ✓ |
| 74 | Test | 1.1505939 | 1.6429378 | 0.700327 | ✓ | ✓ |
| 75 | Test | 0.1318375 | 0.1898685 | 0.694362 | ✗ | ✗ |
| 76 | Test | 0.15018 | 0.2189491 | 0.685913 | ✗ | ✗ |
| 77 | Test | 0.146774 | 0.2144594 | 0.684391 | ✓ | ✓ |
| 78 | Test | 0.1123303 | 0.1665129 | 0.674604 | ✓ | ✓ |
| 79 | Test | 0.1623439 | 0.2468262 | 0.657726 | ✗ | ✗ |
| 80 | Test | 0.4243661 | 0.6563517 | 0.646553 | ✗ | ✗ |
| 81 | Test | 0.2945639 | 0.4662018 | 0.631838 | ✗ | ✓ |
| 82 | Validation | 0.0892761 | 0.1419142 | 0.629085 | ✓ | ✗ |
| 83 | Test | 0.1992316 | 0.3229269 | 0.616956 | ✓ | ✓ |
| 84 | Validation | 0.3260828 | 0.5294719 | 0.615864 | ✓ | ✓ |
| 85 | Test | 0.2181703 | 0.3576818 | 0.609956 | ✓ | ✓ |
| 86 | Test | 0.1757585 | 0.3006565 | 0.584582 | ✓ | ✓ |
| 87 | Validation | 0.1811467 | 0.3107196 | 0.582991 | ✓ | ✓ |
| 88 | Test | 0.2774191 | 0.4759698 | 0.58285 | ✗ | ✓ |
| 89 | Test | 0.137414 | 0.2358583 | 0.582613 | ✓ | ✓ |
| 90 | Validation | 0.1051238 | 0.1834589 | 0.57301 | ✓ | ✓ |
| 91 | Validation | 1.5624891 | 2.7446374 | 0.569288 | ✓ | ✓ |
| 92 | Validation | 0.1104184 | 0.1958337 | 0.563838 | ✗ | ✗ |
| 93 | Validation | 0.1233551 | 0.2228252 | 0.553596 | ✗ | ✗ |
| 94 | Validation | 0.189019 | 0.3445496 | 0.548597 | ✗ | ✗ |
| 95 | Validation | 0.2997031 | 0.5486731 | 0.546233 | ✓ | ✓ |
| 96 | Test | 0.1057559 | 0.19453 | 0.543648 | ✓ | ✓ |
| 97 | Validation | 0.129731 | 0.2426926 | 0.534549 | ✓ | ✓ |

| Task # | Test/Val | PROSE Time (s) | NGDS Time (s) | Speed-up | PROSE Correct? | NGDS Correct? |
|--------|----------|----------------|---------------|----------|----------------|---------------|
| 98  | Test       | 0.1706376 | 0.320323  | 0.532705 | ✓ | ✓ |
| 99  | Test       | 0.0936175 | 0.1764753 | 0.530485 | ✓ | ✓ |
| 100 | Test       | 0.2101397 | 0.40277   | 0.521736 | ✗ | ✗ |
| 101 | Test       | 0.1816704 | 0.3507656 | 0.517925 | ✓ | ✓ |
| 102 | Validation | 0.1516109 | 0.2993282 | 0.506504 | ✓ | ✓ |
| 103 | Test       | 0.1102942 | 0.2185006 | 0.504778 | ✓ | ✓ |
| 104 | Validation | 1.1538661 | 2.3299578 | 0.49523  | ✗ | ✓ |
| 105 | Test       | 0.1241092 | 0.251046  | 0.494368 | ✗ | ✓ |
| 106 | Test       | 1.068263  | 2.176145  | 0.490897 | ✗ | ✗ |
| 107 | Validation | 0.1899474 | 0.389012  | 0.488282 | ✓ | ✓ |
| 108 | Validation | 0.205652  | 0.4312716 | 0.47685  | ✓ | ✗ |
| 109 | Test       | 0.1332348 | 0.2819654 | 0.472522 | ✓ | ✓ |
| 110 | Test       | 0.2137989 | 0.4625152 | 0.462253 | ✗ | ✗ |
| 111 | Validation | 0.2233911 | 0.4898705 | 0.456021 | ✗ | ✗ |
| 112 | Validation | 0.1742123 | 0.3872159 | 0.44991  | ✓ | ✓ |
| 113 | Test       | 0.1798306 | 0.4059525 | 0.442984 | ✓ | ✓ |
| 114 | Validation | 0.1576141 | 0.3592128 | 0.438776 | ✓ | ✓ |
| 115 | Test       | 0.1441545 | 0.3462711 | 0.416305 | ✓ | ✓ |
| 116 | Validation | 0.189833  | 0.4649153 | 0.408317 | ✗ | ✗ |
| 117 | Validation | 0.3401477 | 1.0468088 | 0.324938 | ✓ | ✓ |
| 118 | Validation | 0.1575744 | 0.6015111 | 0.261964 | ✗ | ✗ |
| 119 | Validation | 0.7252624 | 3.2088775 | 0.226017 | ✓ | ✗ |
| 120 | Test       | 0.1288099 | 0.5958986 | 0.216161 | ✓ | ✓ |

## C  ML-BASED RANKER

As noted in Section 2, learning a ranking function is an interesting problem in itself and is orthogonal to our work. Since our method can be used along with any accurate ranking function, we assume black-box access to such a high-quality ranker and specifically, use the state-of-the-art ranking function of PROSE that involves a significant amount of hand engineering.

In this section, we evaluate the performance of our method and PROSE when employing a competitive ranker learned in a data-driven manner (Gulwani & Jain, 2017). From the table below, it can be observed that when using an ML-based ranking function, our method achieves an average $\approx 2\times$ speed-up over PROSE while still achieving comparable generalization accuracy .

| Metric | PROSE | NGDS ($T_1$, BB) | NGDS ($T_1 + POS$, BB) |
|--------|-------|------------------|------------------------|
| **Accuracy (% of 73)** | **65.75** | **65.75** | 64.38 |
| **Speed-up ($\times$ PROSE)** | 1.00 | 2.15 | **2.46** |

Table 5: Generalization accuracy and speed-up of NGDS variants vs. PROSE where all methods use a machine learning based ranking function from Gulwani & Jain (2017).

