# OpenReview forum: "Neural-Guided Deductive Search for Real-Time Program Synthesis from Examples"
_ICLR.cc/2018/Conference — Accept (Poster)_

### Official Review · AnonReviewer2 · 2017-11-24
**Although the search method chosen was reasonable, the only real innovation here is to use the LSTM to learn a search heuristic.**

**Rating:** 6
**Confidence:** 4

**Review:**

The paper presents a branch-and-bound approach to learn good programs
(consistent with data, expected to generalise well), where an LSTM is
used to predict which branches in the search tree should lead to good
programs (at the leaves of the search tree). The LSTM learns from
inputs of program spec + candidate branch (given by a grammar
production rule) and ouputs of quality scores for programms. The issue
of how greedy to be in this search is addressed.

In the authors' set up we simply assume we are given a 'ranking
function' h as an input (which we treat as black-box). In practice
this will simply be a guess (perhaps a good educated one) on which
programs will perform correctly on future data. As the authors
indicate, a more ambitious paper would consider learning h, rather
than assuming it as a given.

The paper has a number of positive features. It is clearly written
(without typo or grammatical problems). The empirical evaluation
against PROSE is properly done and shows the presented method working
as hoped. This was a competent approach to an interesting (real)
problem. However, the 'deep learning' aspect of the paper is not
prominent: an LSTM is used as a plug-in and that is about it. Also,
although the search method chosen was reasonable, the only real
innovation here is to use the LSTM to learn a search heuristic.


The authors do not explain what "without attention" means.


I think the authors should mention the existence of (logic) program
synthesis using inductive logic programming. There are also (closely
related) methods developed by the LOPSTR (logic-based program
synthesis and transformation) community. Many of the ideas here are
reminiscent of methods existing in those communities (e.g. top-down search
with heuristics). The use of a grammar to define the space of programs
is similar to the "DLAB" formalism developed by researchers at KU
Leuven.

ADDED AFTER REVISIONS/DISCUSSIONS

The revised paper has a number of improvements which had led me to give it slightly higher rating.

---

> ### Author Response · Authors · 2017-12-13
> **We combine deep learning & symbolic methods in a new milestone for the program synthesis application, as opposed to a pure deep learning contribution**
>
> Thank you for the related work suggestions -- we will update this discussion in the next draft. We address your concerns below:
>
> > Q: Limited innovation in terms of deep learning:
>
> Rather than being a pure contribution to deep learning, this work applies deep learning to the important field of program synthesis, where statistical approaches are still underexplored. Our main contribution is a hybrid approach to program synthesis that utilizes the best of both neural and symbolic synthesis techniques. Combining insights from both worlds in this way achieves a new milestone in program synthesis performance: from a single example it generates programs that generalize better than prior state-of-the-art (including neural RobustFill, symbolic PROSE, and hybrid DeepCoder), the generated program is provably correct, and the generation is 50% faster on average
>
> DeepCoder (Balog et al., ICLR 2017) first explored a hybrid approach last year by first predicting the likelihood of various operators and then using it to guide an external symbolic synthesis engine. Since deep networks are data-hungry, Balog et al. obtain training data by randomly sampling programs from the DSL and generating satisfying random strings as input-output examples. As noted in Section 1 and as evidenced by its inferior performance against our method, the generated programs tend to be unnatural leading to poor generalization. In contrast, NGDS closely integrates neural models at each step of the synthesis and so, it is possible to obtain large amounts of training data while utilizing a relatively small number of real-world examples.
>
> > Q: Learning the ranking function instead of taking it as a given:
>
> While related, this problem is orthogonal to our work: a ranking function evaluates whether a given full program generalizes well, whereas we aim to predict the generalization of the best program produced from a given partial search state.
>
> Importantly, the proposed technique, NGDS is independent of the ranking function and can be trivially integrated with any high-quality ranking function. For instance, the manually written ranking function of FlashFill in PROSE that we use is a result of 7 years of engineering and heavy fine-tuning for industrial applications. An even better-quality learned ranking function would only improve the accuracy of predictions, which are already on par with baseline PROSE (68.49% vs 67.12%).
>
> In fact, a lot of recent prior work focuses on learning a ranking function for program induction, see (Singh & Gulwani, CAV 2015) and (Ellis & Gulwani, IJCAI 2017). For comparison, we are currently performing a set of experiments with an ML-learned ranking function; we'll update with the new results once it's done.
>
> > Q: What does "without attention" mean?
>
> All the models we explore encode input and output examples using (possibly multi-layered, bi-directional) LSTMs with or without an attention mechanism (Bahdanau et al., ICLR 2015). As mentioned in Section 8, the most accurate predictions arise when we attend to the input string while encoding the output string similar to the attention-based models proposed by Devlin et al., 2017. We will make this clearer in the next version of the paper.
>
> Such an attention mechanism allows the network to learn complex features like "whether the output is a substring of the input". Unfortunately, such accuracy comes at a cost of increasing the network evaluation time to quadratic instead of linear. As a result, prediction time at every node of the search tree dominates the search time, and NGDS is slower than PROSE even when its predictions are accurate. Therefore, we only use LSTM models without any attention mechanism in our evaluations.

---

### Official Review · AnonReviewer3 · 2017-11-27
**Incremental paper but well-written**

**Rating:** 6
**Confidence:** 3

**Review:**

This paper extends and speeds up PROSE, a programming by example system, by posing the selection of the next production rule in the grammar as a supervised learning problem.

This paper requires a large amount of background knowledge as it depends on understanding program synthesis as it is done in the programming languages community. Moreover the work mentions a neurally-guided search, but little time is spent on that portion of their contribution. I am not even clear how their system is trained.

The experimental results do show the programs can be faster but only if the user is willing to suffer a loss in accuracy. It is difficult to conclude overall if the technique helps in synthesis.

---

> ### Author Response · Authors · 2017-12-13
> **Training details + Clarification on the generalization accuracy**
>
> > Q: Please clarify how the system is trained.
>
> 1) We use the industrially collected set of 375 string transformation tasks. Each task is a single input-output examples and 2-10 unseen inputs for evaluating generalization. Further, we split the 375 tasks into 65% train, 15% validation, and 20% test ones.
> 2) We run PROSE on each of those tasks and collect the (symbol, production, spec input, spec output -> best program score after learning) information on all nodes of the search tree. As mentioned in the introduction, such traces provide a rich description of the synthesis problem thanks to the Markovian nature of deductive search in PROSE and enabling the creation of large datasets required for learning deep models. As a result, we obtain a dataset of ~450,000 search outcomes from mere 375 tasks.
> 3) We further split all the search outcomes by the used symbol or its depth in the grammar. In our final evaluation, we present the results for the models trained on the decisions on the `transform` (depth=1), `pp`, `pos` symbols. We have also trained other symbol models as well as a single common model for all symbols/depths, but they didn’t perform as well.
> 4) We employ Adam (Kingma and Ba, 2014) to optimize the objective. We use a batch size of 32 and a learning rate of 0.01 and use early stopping to pick the final model. The model architecture and the corresponding loss function (squared error) are discussed in Section 3.1. We will add the specific training details in the next revision of the paper.
> 5) As discussed in Section 3.3, the learned models are integrated in the corresponding PROSE controller when the current search tree node matches the model's conditions (i.e. it is on the same respective symbol or depth).
>
> > Q: Is the approach useful for synthesis when there is a loss in program accuracy?
>
> In fact, NGDS achieves higher average test accuracy than baseline PROSE (68.49% vs. 67.12%), although with slightly lower validation accuracy (63.83% vs. 70.21%) which effectively corresponds to 4 tasks.
>
> However, this is not the most important factor: PBE is bound to often fail in synthesizing the _intended_ program from a single input-output example. Even a machine-learned ranking function picks the wrong program 20% of the time (Ellis & Gulwani, IJCAI 2017).
>
> Thus, the main goal of this work is speeding up the synthesis process on difficult scenarios without sacrificing the generalization accuracy too much. As a result, we achieve on average 50% faster synthesis time, with 10x speed-ups for many difficult tasks that require multiple seconds while still retaining competitive accuracy. Appendix C shows the breakdown of time and accuracy: out of 120 validation/test tasks, there are:
> - 76 tasks where both systems are correct,
> - 7 tasks where PROSE learns a correct program and NGDS learns a wrong one,
> - 4 tasks where PROSE learns a wrong program and NGDS learns a correct one,
> - 33 tasks where both systems are wrong.

---

### Official Review · AnonReviewer1 · 2017-12-01
**Strong paper; accept**

**Rating:** 8
**Confidence:** 3

**Review:**

This is a strong paper. It focuses on an important problem (speeding up program synthesis), it’s generally very well-written, and it features thorough evaluation. The results are impressive: the proposed system synthesizes programs from a single example that generalize better than prior state-of-the-art, and it does so ~50% faster on average.

In Appendix C, for over half of the tasks, NGDS is slower than PROSE (by up to a factor of 20, in the worst case). What types of tasks are these? In the results, you highlight a couple of specific cases where NGDS is significantly *faster* than PROSE—I would like to see some analysis of the cases were it is slower, as well. I do recognize that in all of these cases, PROSE is already quite fast (less than 1 second, often much less) so these large relative slowdowns likely don’t lead to a noticeable absolute difference in speed. Still, it would be nice to know what is going on here.

Overall, this is a strong paper, and I would advocate for accepting it.


A few more specific comments:


Page 2, “Neural-Guided Deductive Search” paragraph: use of the word “imbibes” - while technically accurate, this use doesn’t reflect the most common usage of the word (“to drink”). I found it very jarring.

The paper is very well-written overall, but I found the introduction to be unsatisfyingly vague—it was hard for me to evaluate your “key observations” when I couldn’t quite yet tell what the system you’re proposing actually does. The paragraph about “key observation III” finally reveals some of these details—I would suggest moving this much earlier in the introduction.

Page 4, “Appendix A shows the resulting search DAG” - As this is a figure accompanying a specific illustrative example, it belongs in this section, rather than forcing the reader to hunt for it in the Appendix.

---

> ### Author Response · Authors · 2017-12-13
> **Error analysis**
>
> Thank you for the constructive feedback! We’ll add more details and clarify the introduction in the next revision.
>
> Q: Which factors lead to NGDS being slower than PROSE on some tasks?
> Our method is slower than PROSE when the predictions do not satisfy the requirements of the controller i.e. all the predicted scores are within the threshold or they violate the actual scores in branch and bound exploration. This leads to NGDS evaluating the LSTM for branches that were previously pruned. This can be especially harmful when branches that got pruned out at the very beginning of the search need to be reconsidered -- as it could lead to evaluating the network many times. While evaluating the network leads to minor additions in run-time, there are many such additions, and since PROSE performance is already << 1s for such cases, this results in considerable relative slowdown.
>
> Why do the predictions violate the controller's requirements? This happens when the neural network is either indecisive (its predicted scores for all branches are too close) or wrong (its predicted scores have exactly the opposite order of the actual program scores).
> We will update the draft with this discussion and present some examples below
>
> Some examples:
> A) "41.711483001709,-91.4123382568359,41.6076278686523,-91.6373901367188"  ==>  "41.711483001709"
> 	The intended program is a simple substring extraction. However, at depth 1, the predicted score of Concat is much higher than the predicted score of Atom, and thus we end up exploring only the Concat branch. The found Concat program is incorrect because it uses absolute position indexes and does not generalize to other similar extraction tasks with different floating-point values in the input strings.
> We found this scenario relatively common when the output string contains punctuation - the model considers it a strong signal for Concat.
> B) "type size =  36: Bartok.Analysis.CallGraphNode type size =  32: Bartok.Analysis.CallGraphNode CallGraphNode"  ==> "36->32"
> 	We correctly explore only the Concat branch, but the slowdown happens at the level of the `pos` symbol. There are many different logics to extract the “36” and “32” substrings. NGDS explores RelativePosition branch first, but the score of the resulting program is less then the prediction for RegexPositionRelative. Thus, the B&B controller explores both branches anyway and we end up with a relative slowdown caused by the network inference time.

---

### Author Response · Authors · 2017-12-21
**New revision**

We have uploaded a new paper revision, as per the reviewers' feedback. Here's a summary of the changes:

- Restructured the introduction, making NGDS details clearer and moving them earlier.
- Added analysis of some erroneous scenarios in the Evaluation.
- Expanded related work overview with more symbolic methods such as ILP and LOPSTR.
- Added details of the training process, including all the hyperparameters.
- Moved Appendix A into the main text.
- Replaced the table in Appendix B (earlier C): we found that we selected a wrong model to generate the table in the previous submission. The summary results in Tables 1-2 and their analysis were on the correct best model (so no change was needed in the Evaluation), but the spreadsheet for detailed results in the appendix was not. We apologize for this confusion. The distribution of the speed-ups did not change substantially, although the correct spread is now from 12x to 0.2x.

We will upload one more revision later this month, in which we'll include experiments we're currently performing with an ML-learned ranking function (as opposed to the state-of-the-art PROSE ranking function, used in the current submission).

---

> ### Author Response · Authors · 2018-01-03
> **revision with ML-based ranker**
>
> Following reviewers' feedback, we have updated the draft (Appendix C) with experiments that employ an ML-based ranking function as against the state-of-the-art ranker of PROSE that involves hand engineering. We observe that NGDS achieves ~2X speed-ups on average while still achieving highly comparable generalization accuracy as compared to PROSE with the ML-based ranker.

---

### Decision · Program_Chairs · 2018-01-29
**ICLR 2018 Conference Acceptance Decision**

**Decision:**

Accept (Poster)

**Comment:**

The pros and cons of this paper cited by the reviewers can be summarized below:

Pros:
* The method proposed here is highly technically sophisticated and appropriate for the problem of program synthesis from examples
* The results are convincing, demonstrating that the proposed method is able to greatly speed up search in an existing synthesis system

Cons:
* The contribution in terms of machine learning or representation learning is minimal (mainly adding an LSTM to an existing system)
* The overall system itself is quite complicated, which might raise the barrier of entry to other researchers who might want to follow the work, limiting impact

In our decision, the fact that the paper significantly moves forward the state of the art in this area outweighs the concerns about lack of machine learning contribution or barrier of entry.